# Neurotoxicity in Marine Invertebrates: An Update

**DOI:** 10.3390/biology10020161

**Published:** 2021-02-18

**Authors:** Irene Deidda, Roberta Russo, Rosa Bonaventura, Caterina Costa, Francesca Zito, Nadia Lampiasi

**Affiliations:** Istituto per la Ricerca e l’Innovazione Biomedica IRIB, Consiglio Nazionale delle Ricerche, Via Ugo La Malfa 153, 90146 Palermo, Italy; irene.deidda@irib.cnr.it (I.D.); roberta.russo@irib.cnr.it (R.R.); rosa.bonaventura@irib.cnr.it (R.B.); caterina.costa@irib.cnr.it (C.C.); francesca.zito@irib.cnr.it (F.Z.)

**Keywords:** nervous system, echinoderms, molluscs, crustaceans, emerging contaminants, heavy metals, pesticides, microplastics, omics

## Abstract

**Simple Summary:**

The pollution of air, soil, and sea has grown in recent decades at the same pace as human development. Climate changes add further damage to the ecosystems. Nowadays, pollutants that derive from the anthropization of the environment are indicated as “emerging pollutants”. Marine organisms, especially invertebrates, are used as model systems for ecotoxicological studies also regarding the nervous system, even if studies on pollutants’ neurotoxic effects are still few. A great leap forward in knowledge can come from integrated omics studies that bring together genomics, transcriptomics, proteomics, and metabolomics data. These studies have revealed that pollutants are dangerous for the life of marine organisms, and not only because they can be modified in the environment, but also because they can combine giving rise to new mixtures. These new combinations can be even more harmful than individual pollutants. We must not forget that many marine organisms, both invertebrates and vertebrates, become part of the human food chain. Therefore, ultimately, the pollutants that contaminate the air, soil, and sea are potentially harmful to human health. The purpose of this review is to focus on some of the pollutants that are most frequently present in marine environments and analyze their neurotoxic potential effects on some marine model organisms.

**Abstract:**

Invertebrates represent about 95% of existing species, and most of them belong to aquatic ecosystems. Marine invertebrates are found at intermediate levels of the food chain and, therefore, they play a central role in the biodiversity of ecosystems. Furthermore, these organisms have a short life cycle, easy laboratory manipulation, and high sensitivity to marine pollution and, therefore, they are considered to be optimal bioindicators for assessing detrimental chemical agents that are related to the marine environment and with potential toxicity to human health, including neurotoxicity. In general, albeit simple, the nervous system of marine invertebrates is composed of neuronal and glial cells, and it exhibits biochemical and functional similarities with the vertebrate nervous system, including humans. In recent decades, new genetic and transcriptomic technologies have made the identification of many neural genes and transcription factors homologous to those in humans possible. Neuroinflammation, oxidative stress, and altered levels of neurotransmitters are some of the aspects of neurotoxic effects that can also occur in marine invertebrate organisms. The purpose of this review is to provide an overview of major marine pollutants, such as heavy metals, pesticides, and micro and nano-plastics, with a focus on their neurotoxic effects in marine invertebrate organisms. This review could be a stimulus to bio-research towards the use of invertebrate model systems other than traditional, ethically questionable, time-consuming, and highly expensive mammalian models.

## 1. Introduction

In the frame of the global environmental changes, rising and worrying data on both the environmental contamination and the related toxicology have been recorded since the mid-twentieth century, when the human activities that are oriented towards industrialization and globalization began to intensify. In particular, marine ecosystems exhibit a wide variety of hazardous anthropogenic pollutants that pervade the entire water column, including sediments, especially along the coasts that exhibit the different levels of human activities releasing industrial, agricultural, hospitals, clinics, and domestic wastes into the sea [1,2,3]. Without considering the biodegradable contaminants that arise from organic waste, the main global concerns are related to the release of emerging contaminants (ECs), also called micropollutants, including, among others, heavy metals, microplastics, nanoparticles, pharmaceuticals, and personal care products (PPCPs) (drugs chemicals, etc.), xenobiotics, such as pesticides (insecticides, herbicides, and fungicides), and endocrine disrupting compounds (EDC), etc. Due to their persistence in the environment, accumulation in the living tissues via the food chains, and toxicity, they endanger the health of the marine biota as well as the human health [4,5,6,7]. The ECs impact can determine sub-lethal, acute, or chronic effects to the single organism level, or can be extended to the population level, influencing, in this case, population dynamics, genetic, and diversity of the ecosystem [2]. Through the impairment of cellular homeostasis and gene expression regulation, the ECs can cause human disorders, as well as determine epigenetic phenomena altering gene functions with inherited effects [8]. Therefore, many ECs, even if in trace, can represent a high environmental risk factor, reached by trophic transfer animal species, and cause several adverse effects in the human health, including alterations in the structure and function of the nervous system (NS) and, thus, contribute to neurodevelopmental disorders (DNT) and neurological diseases. Most neurotoxic contaminants are highly lipophilic and they easily cross the blood brain barrier (BBB) and the placenta [4] (Figure 1).

The environmental risk assessment is currently based on laboratory, mesocosms, and in the field researches, while taking in account that each single chemical introduced in the environment can undergo degradation or interact with other chemicals, or natural factors (biotic/abiotic), leading to synergistic or antagonistic effects [3,9,10]. Although growing evidence points to the importance of considering interactions between stressors in the environment, insufficient literature has been produced so far [2,10,11,12].

Marine invertebrates are widely utilized as strategic model systems for their easy maintenance and handling, small life cycle, and simplified structures showing a major plasticity as compared to those of vertebrates. Adults and planktonic larval stages of ecologically relevant benthic species from Echinodermata, Arthropoda (crustacea), and Mollusca phyla have been extensively used in both environmental risk assessment and to test pollutants toxic effects, including neurotoxic effects, since they exhibit a well-developed, albeit simplified, NS [13,14,15,16]. Many species of crustacean decapods have been utilized over the past decades as biomedical models to understand neural network and physiology due to neurons of large size and relatively simple nervous circuits. Recent high-throughput approaches, as new generation genoma and transcriptoma analysis, made it possible to identify molecules that are involved in the neuromodulator/neurotransmitter pathways and in their control [17].

Neurotoxic effects have been described in several organisms, including marine invertebrates, through exposure to different pollutants in conventional ecotoxicological assays. The main toxic and neurotoxic mechanisms manifested turn out to be Ca2+ dysregulation, oxidative stress with reactive oxygen species (ROS) production, DNA damage, lipid peroxidation (LPO), histopathological damage, apoptosis, mitochondrial dysfunctions, neuro-inflammation, altered dendritic growth and neuronal connectivity, neuronal apoptosis, and developmental neurotoxicity [7].

Recently, new high-throughput approaches from the era of post-genomic and second-generation sequencing have developed the “systems toxicology”, combining toxic-genomic, proteomic and metabolomic fields, in order to simultaneously analyze the differential expression of genes (DEG), proteins, and metabolites, and to find correlations between the expression levels of specific biomarkers and exposure to chemicals [9,18,19]. Of great importance is the combination between specific phenotypes generated by environmental toxic factors and system toxicology integrated data in order to understand response mechanisms in different species and identify advanced ecotoxicological diagnostic tools for refining the ecological hazard assessment process and improving the assessments in the environmental bio-monitoring [20]. In this context, bioinformatic and mathematical models assume great importance in predicting and providing mechanistic explanations on mixture toxicity [11,12].

In this review, divided in three sections, we focus on: (1) the acetylcholine neurotrasmitter and the acetylcholinesterase as biomarkers of neurotoxicity, (2) the scientific literature on some neurotoxic ECs (heavy metals, pesticides, microplastics) and some studies on their effects on adults and developmental stages of some species from Echinodermata, Arthropoda (crustacea) and Mollusca phyla, and (3) the recent omics studies highlighting new molecular pathways that are affected by marine pollution.

This review, which highlights the importance of multidisciplinary and integrated studies, aims to stimulate investigations in this direction, in order to better understand the ecosystem global response and contribute to urgent policy decisions on the environment and human health protection from ECs.

## 2. Acetylcholinesterase as Principal Biomarker of Neurotoxicity

Acetylcholine (ACh) is an excitatory cholinergic neurotransmitter that is present in both vertebrates and many invertebrates that principally regulates the neuromuscular transmission. ACh is synthesized by the choline acetyltransferase (ChAT) enzyme from acetyl coenzyme A and choline in the nerve terminals. In vertebrates, ACh acts as neurotransmitter in both the parasympathetic and sympathetic nervous systems [21]. In invertebrate organisms, the neuromuscular transmission is both glutamatergic and cholinergic. ACh mediates the longitudinal muscle contraction in echinoderms (intestinal muscle of holothurians, esophagus of sea urchin, and cardiac stomach of sea star); the stomatogastric system in the crustaceans is principally cholinergic and it controls visceral motor activity; in molluscs and annelids, cholinergic neurons regulate the gut motility, although several other neurotransmitters and neuropeptides are involved [22,23].

Cholinesterases (ChEs) are a group of enzymes that are absent in the plant and present in many multicellular organisms up to vertebrates that have a key role in the nervous system [24]. Depending on the species, there are different genes encoding cholinesterase enzyme and diverse polymorphic variants [25]. The two principal classes of ChE are acetylcholinesterase (AChE) and butyrylcholinesterase (BuChE), which have been well characterized through their substrate specificity, response to selective inhibitors, and kinetic parameters [26].

AChE is an essential enzyme in the cholinergic nervous system of both vertebrates and invertebrates, whose principal physiological function is the hydrolysis of the neurotransmitter ACh and its removal from the synaptic cleft or blood plasma. Furthermore, AChE is the highly sensitive target for several chemical agents, such as organophosphates or carbamates pesticides, which are responsible for its inhibition. The inactivation of AChE causes ACh accumulation in synapses and continuous stimulation of cholinergic receptors, thus leading to an alteration of neurotransmission and paralysis. For this reason, the inhibition of its activity is considered as a common biomarker of pesticide exposure and, in general, as a sensor of neurotoxicity [26]. In particular, this occurs in vertebrate species in which AChE has been identified in erythroid cells and in several tissues, such as brain, skeletal muscle, liver, kidney, and lungs, and where it is involved in cell development, inflammation, apoptosis, and neuronal development and regeneration [27].

The AChE inhibition as a biomarker has also been considered in marine invertebrates, such as molluscs, nematodes, crustaceans, and echinoderms, analyzing whole-body tissues or specific tissues, such as haemolymph, gills, mantle, digestive glands, and eyes [21,26,28]. AChE activity levels, for example, in oysters, is used as a biomarker of exposure to neurotoxic compounds [21]. In addition to technical variables and species-specific differences, the variability in the sensitivity of ChE is given by the different forms that are present in the same species and the use of non-specific neurological tissues. In some reports, including invertebrate species, AChE showed high sensitivity to toxic compounds with properties that are similar to vertebrates, also correlating its levels with the rate of the mortality.

## 3. Marine Pollutants: Heavy Metals

Metals are natural constituents of the Earth’s crust present in all ecosystems at variable concentrations. The use of metals has accompanied civilization since prehistoric times, characterizing the Bronze and Iron Ages, the Industrial Revolution, up to the present modern society, where they are closely linked to anthropogenic activities, including agriculture, transport, arts, and medicine [29,30]. Metals are found in the wires, batteries, and microchips, as well as in many common everyday objects. Although some heavy metals are crucial for life, their excess can be toxic and therefore extreme attention is paid to the possible consequences for human health and environment [29]. Because of their accumulation in the environment and their significant toxicity, many metals and metalloids have been identified as “priority substances” and included in the list of the Annex II of the European Directive 2008/105/EC (Annex X, Water Framework Directive 2000, https://echa.europa.eu/ann-x-priority-subs-water-dir-2000-60 (accessed on 20 July 2020)). In this regard, several metals, such as arsenic, lead, and mercury, have been included in the 2019 Substance Priority List by the Agency for Toxic Substances and Disease Registry (ATSDR) (https://www.atsdr.cdc.gov/spl/ (accessed on 20 July 2020)). Based on their chemical characteristics, the concentrations of metals are variable in the marine system and some metals tend to accumulate in the bottom sediments. Numerous species of marine invertebrates hyperaccumulate metals (such as arsenic, copper, iron, and zinc), thus resulting in significant relevance to human health and aquatic toxicity [31]. In depth analyses on the speciation, bioavailability, and bioaccumulation of some metals have been well described in other reviews [31,32,33].

Although many of the harmful effects that are induced by exposure of some heavy metals are well known, the mechanisms underlying their toxicity in humans have not yet been fully clarified. Numerous studies mainly conducted on fishes and mammals have highlighted the close correlation between exposure of heavy metals and neuronal damage [29,34,35]. Many of heavy metals can cross the blood brain barrier (BBB) and accumulate in the nervous tissues, inducing oxidative stress, inflammation, and metabolic dysfunction linked to the development of several neurodegenerative diseases.

Albeit the marine organisms are ideal indicators for long-term monitoring of metal accumulation [32], only few studies have been aimed to better understand the neurotoxic effects of heavy metals in marine invertebrates. Through in vitro and in vivo experiments, most of these studies have been focused on the analysis of the AChE activity as a possible biomarker of neurotoxicity in response to excess metals. Some studies have analyzed the combined neurotoxic effects of the different metals that are present in the polluted areas, while very few others have investigated the mechanisms underlying the toxicity that is induced by these substances at the neuronal level, including sensory alterations and neuromuscular dysfunctions.

Here, we focused our attention on neurotoxic effects that are caused by the exposure of some metals (see Table 1) in some marine invertebrates also having key positions in food chains.

Table 1 shows the classification of neurotoxic metals and metalloids in essential and not essential (modified from Caito and Aschner 2015). *Metalloid.

### 3.1. Essential Metals

Among the essential heavy metals, Manganese (Mn) is one of the most abundant elements in the Earth’s crust and a cofactor of numerous metabolic and antioxidant enzymes that are implicated in several biological processes, including cellular homeostasis, immunity, growth, and development [36]. Moreover, Mn plays a key role in the development and functioning of the nervous system, and its levels are tightly regulated, as its excess causes motor and cognitive alterations in humans. The neurons most sensitive and most affected by the accumulation of Mn are the dopaminergic neurons that trigger manganism syndrome with clinical features that are similar to Parkinson’s disease (PD) [37]. To date, the principal mechanism of Mn neurotoxicity is not completely understood.

Among marine invertebrates, the crustacean Norway lobster (*N. norvegicus*) has been mainly used to evaluate the neurotoxic effects that are caused by excessive exposure of Mn through both experimental and field studies [38,39]. It has been hypothesized that the Mn excess can affect the neuromuscular system of this organism through an inhibitory effect on calcium-regulated ion channels [40]. Moreover, the exposure of *N. norvegicus* to Mn (90 and 180 µM) for three weeks resulted in the identification of the brain as the principal tissue of Mn accumulation. High levels were also found in another nervous tissue, the abdominal ganglia, whereas haemolymph and muscle tissue showed lower concentrations of Mn. In this study, several neuromuscular functions were also evaluated (free swimming performance and the force measurement in tethered animals) and only a significant reduction was found in the muscle performance, which was negatively correlated with the Mn concentration present in the abdominal ganglia [38]. Mn accumulation was also analyzed in the American lobster (*H. americanus*) following acute (96 h) and long-term (seven weeks) Mn exposure at different concentrations (20, 80, 150, and 300 mg/L). In all tissues examined, included optic nerve and brain, Mn accumulation was found in a dose-dependent manner. Although a possible association between exposure to high concentration of Mn and blindness was observed, this was not significant [41].

Alteration in chemoreceptive mechanism was also hypothesized in an experimental study regarding swimming (food search) behavior in which *N. norvegicus* was exposed to Mn (0.1 and 0.2 mM) for 12 days. The results obtained showed an increase of reaction time to food odor stimuli and a reduced number of lobsters able to reach the food source [42].

Other marine organisms, belonging to the Mollusca phylum, have been used to test Mn neurotoxicity. In particular, the eastern oyster *Crassostrea virginica* was used to study the effects of Mn in the dopaminergic system. The exposure of *C. virginica* to Mn at two different doses (0.5 and 1 mM) for three days resulted in a significant decrease of the dopamine levels in the cerebral and visceral ganglia and in the gill. No changes were detected in the levels of serotonin, the neurotransmitter that, together with dopamine, regulates the beating rates of the cilia in the nervous tissues (cerebral and visceral ganglia) and gill of oyster. Moreover, exposure of oyster’s gill to Mn resulted in the alteration of dopaminergic cilio-inhibitory system, even after exogenous application of dopamine, thus suggesting a direct effect of Mn primarily in post-synaptic sites [43]. The same results were obtained by another study that was conducted in the same oyster species, in which the protective effects of p-aminosalicylic acid (PAS) against Mn-induced neurotoxicity were demonstrated [44]. To better understand the mechanisms underlying Mn neurotoxicity, the dopamine D2-like receptor (D2R) signal pathway, responsible for inhibiting adenyl cyclase and slowing beating rates of lateral cilia in the gill, was examined. Using an activator or two different inhibitors of adenylyl cyclase, the authors identified the D2R receptor as a possible target of Mn-toxicity that was able to directly inhibit its activation or act on downstream signaling activity without affecting adenylyl cyclase [45].

Few studies have been focused on Mn toxicity in echinoderms, which exhibit a lower sensitivity to this metal compared to crustaceans and molluscs. A study has investigated the neurotoxic effects of Mn in two asteroid echinoderms, the common sea star, *Asterias rubens,* and the black brittle star, *Ophiocomina nigra*. In particular, the exposure of both species to Mn (12 mg/L) for 12 days resulted in a motor dysfunction that was analyzed by the turning-over assay (measuring the ability to move from ventral to dorsal position). This alteration was not heal after two weeks of recovery. Although a small amount of Mn was found in the digestive organ and the pyloric caeca, the tube feet, namely the sensory structures also used in the adhesion and locomotion processes, were identified as the principal organ of Mn accumulation. Further evidence of the alteration in the neuronal functions caused by Mn exposure was the increase of the AChE activity that was detected in the tube feet in which the levels of Mn also remained high after the recovery period [46].

To date, very few studies have addressed the neurotoxic effects in marine invertebrates that are caused by exposure of the other two essential metals, iron and zinc, and, for this reason, we treated these two elements together.

Iron (Fe) and Zinc (Zn) are essential elements for life, being involved in several fundamental biological processes in all living organisms. In humans, iron regulates many CNS processes, including the metabolism of neurotransmitters (dopamine, norepinephrine, and serotonin) and myelin [47,48]. Iron excess is toxic inasmuch via the Fenton reaction leads to the release of ROS, bringing about oxidative stress and, consequently, LPO, protein aggregation, and DNA damage [49]. Alterations in Fe and Zn homeostasis have been implicated in the pathogenesis of several CNS neurodegenerative diseases, such as amyotrophic lateral sclerosis (ALS), Friedreich’s ataxia, PD and Alzheimer disease (AD), and vascular dementia [50,51,52].

Despite being abundant elements in the Earth’s crust, the essential metals Fe and Zn are present at low concentrations in the marine environment [53,54]. Fe has a reduced solubility and it is normally found complexed in organic ligands in the water. In marine systems, higher concentrations of Fe are often associated to polluted sources, whereas the high levels for Zn are usually derived from the welling of zinc-rich deep-ocean water [32].

The toxic effects of several metals, including Fe and Zn, were analyzed in two mussel species, *P. perna* and *M. galloprovincialis*, in which the AChE enzyme was characterized by analyzing its activity and distribution in diverse tissues. The highest value of AChE activity was identified in the hepatopancreas and gills in *M. galloprovincialis*, or in hepatopancreas and muscle in *P. perna*, with minimal values also in the gills. Moreover, seasonal variations in the AChE activity have been observed in *P. perna*, which are probably related to sexual rest and spawning. A significant reduction of AChE activity was observed in specimens of the same species collected from polluted sites. In addition, in vitro experiments were conducted to evaluate AChE activity following exposure to several metals. In particular, the whole extracts that were derived from both mussel species were exposed to Fe (FeCl_3_, 10^−2^ and 10^−4^ M), Zn, and copper (ZnCl_2_ and CuSO_4_; 10^−2^–10^−7^ M) as essential metals and cadmium (Cd), as non-essential metal (CdCl_2_, 10^−2^–10^−7^ M) for 30 min. A significant inhibition of AChE activity was observed by all metals and in both mussel species. Exposure to Zn as well as Cd at 10^−3^ and 10^−2^ M caused a significant decrease of AChE activity, whereas Fe and Cu exposure determined a complete inhibition of activity, showing a greater neurotoxic effect when compared to other metals under these experimental conditions. Finally, the in vivo exposure of *P. perna* to Cd (10 μg/L) and Zn (1 mg/L) for 11 days resulted in the inhibition of AChE activity only at one day of exposure for both metals, maintaining an increased level of activity as compared to the control group in the following exposures [55].

The exposure of sea urchin embryos to Zn resulted in the shift of the fate of cells toward the ectodermal tissue from which the nervous system originates, thus expanding the region of the apical sensory ciliary and reducing the endomesodermal tissues [56,57,58]. *SpOnecut* gene expression was analyzed in zinc treated *Strongylocentrotus purpuratus* embryos, and its expression was found to be more expanded after zinc treatment [59]. In mammals, onecut genes are essential genes during early and later development of nervous system [60].

Copper (Cu) is another vital element acting as a cofactor for many key proteins, such as cytochrome c oxidase, ferroxidase ceruloplasmin, and copper enzyme superoxide dismutase. Many marine organisms have the ability to regulate the Cu concentrations in their tissues and to hyperaccumulate it, such as several species of fishes, cephalopod, and bivalve molluscs. In some species of crustaceans and molluscs or in some of their life stage, the high concentrations of Cu found must be carefully considered, because this metal is present in the haemocyanin protein, which is used for respiration. The Cu bioavailability and effects in aquatic organisms have been extensively investigated [31]. However, elevated Cu concentrations can be toxic for aquatic organisms as well as humans, where its accumulation also mediates neurotoxicity. Several multifactorial mechanisms of Cu neurotoxicity have been proposed, suggesting its role in autism spectrum disorders (ASD), a group of neurodevelopmental disorders, and in neurodegenerative diseases, including AD, PD, ALS, and Huntington’s Disease [61,62].

In addition to the previous research [55], other studies have investigated the toxic effects of Cu on the nervous system of molluscs, which are commonly used as bio-indicators of metal pollution. Two different species of marine gasteropods, *Monodonta lineata* and *Nucella lapillus*, were exposed to several concentrations of Cu (CuSO₄·5H₂O; 0.011–0.120 mg/L Cu^2+^) and Cd (CdCl_2_ · H_2_O; 0.950–10 mg/L Cd^2+^) for 96 h evaluating the rate of mortality. Following the characterization of the soluble forms of foot muscle ChE enzyme, the authors analyzed ChE activity after in vitro and in vivo exposures to a range of concentrations of Cu^2+^ and Cd^2+^ (for in vitro experiments: 3.1–100 mg/L; for in vivo experiments: Cu^2+^, 0.016–0.060 mg/L for *M. lineata* and 0.012–0.044 for *N. lapillus*). The in vitro exposure to Cu of both gastropods species resulted in the decrease of ChE activity (70% of inhibition), whereas no in vivo effects of Cu were observed. No changes in the AChE activity were observed following the in vitro exposure to Cd in both examined species, whereas an increase of AChE activity was only detected in the *N. lapillus* following its in vivo exposure to Cd [63].

The exposure of blue mussel *M. galloprovincialis* to Cu (CuSO_4_, 10^−3^–10^−6^ M; CuCl_2_, 5 × 10^−5^ M) for 40 min resulted in the decrease of total ChE activity in vitro (isolated from whole body). ChE activity was totally inhibited following CuCl_2_ and CuSO_4_ exposure at the highest concentration tested [64].

Several studies aimed to evaluate the possible relationship between exposure to Cu and alteration in the opening/closure of the shell or in the heart rates, but only a few have suggested the nervous system as a target for Cu. This occurred in the study in which the exposure of *M. edulis* to Cu (CuCl_2_, 0.08–12.5 µM) for two days resulted in changes in cardiac activity, not directly correlated to valve closure, but rather related with negative effects on the cardiac neuromuscular system [65].

In addition to *M. edulis*, the toxic effects of Cu were evaluated in two other marine invertebrates, the common limpet *Patella vulgata* (molluscs) and shore crab *Carcinus maenas* (crustaceans). The exposure to Cu (CuCl_2_·2H_2_O; 6.1, 38.5, and 68.1 µg/L) for seven days resulted in the alteration of several biomarkers, including lysosomal stability (neutral red retention), metabolic impairment (total haemolymph protein), physiological status (heart rate), and neurotoxicity (AChE). *P. vulgata* was the species with greater sensitivity to Cu, showing alterations in all biomarkers, starting from 6.1 µg/L of Cu, followed by *C. maenas* and *M. edulis*; this latter species was the most resistant to Cu exposure exhibiting biomarker changes only at the higher concentration, without any change on the AChE activity. The exposure of *P. vulgata* to Cu at 6.1 µg/L resulted in the increase of haemolymph AChE activity. In *C. maenas*, the exposure to Cu at 68.1 µg/L caused a significant reduction of AChE activity and the increase of metallothionein (MT) expression, a protective non-enzymatic metalloprotein that is produced in response of different insults [66]. MT is a metal chelanting protein with detoxifying activity and its expression has been correlated with several metals, including copper, zinc, and cadmium [67,68,69].

Similar results were obtained for the same organism, *C. maenas*, after in vivo exposure to Cu (200 μg/L) for different times (0.5–7 days). A significant inhibition of AChE activity was observed in the digestive gland and in the eye at the early times of Cu exposure, whereas, in the muscle, only after seven days of exposure. In addition, the authors investigated the effects of exposure to Cd (200 μg/L) or to the mixture of Cd + Cu (200 μg/L Cd + 200 μg/L Cu) on the AChE activity, observing a general reduction in all of the examined tissues [70].

Additionally, the in vivo exposure of *C. maenas* to Cd (87.5 and 15 mg/L) for 96 h resulted in the inhibition of haemolymph AChE activity. Similar effects were observed following exposure to Cd and mixtures of both metals [71].

Among marine crustaceans, the marbled crab (*Pachygrapusus marmoratus*), at the larval stage, was exposed to different concentrations of Cu (CuSO_4_·5H_2_O; 0.25–8 mg/L Cu^2+^) for 24 and 48 h and the rate of mortality showed a dose-dependent trend. The calculation of LC_50_ value (0.5 mg/L of Cu^2+^) was used to choose the conditions of in vitro exposure to Cu (0.5, 0.25 and 0.125 mg/L of Cu^2+^ for 1 h), evaluating oxidative stress and neurotoxic effects. The increase of superoxide dismutase (SOD) and catalase (CAT) activities and the reduction of glutathione reductase (GR) activity were observed, starting from the lowest concentration used (0.25 mg/L). The neurotoxic effects were evaluated by analyses of different cholinesterases (AChE, BChE, PChE) and no significant variation of AChE activity was observed when compared to control group. On the contrary, a dose-dependent increase of BChE and PChE activities was reveled [72].

### 3.2. Non-Essential Metals

Cadmium (Cd) and lead (Pb) are non-essential heavy metals and the two most abundant toxic environmental pollutants whose toxicity has been extensively studied [34].

Pb is ubiquitously present in aquatic environments and high levels are associated with anthropogenic activities that are related to products, such as batteries, pigments, and petrol additives [73]. Human exposure to Pb is usually via ingestion or inhalation, and it can affect liver, kidney, and the hematopoietic and cardiovascular systems [62]. The Pb can bio-accumulate in mammals and also in fish, inducing oxidative stress as well as immuno- and neuro-toxicity [74]. The main target of Pb is the nervous system and its exposure leads to cognitive and behavioral deficits [75].

Diet and smoke are the major routes of human exposure of Cd. Besides having teratogenic and mutagenic effects, Cd has been classified as a human carcinogen. Chronic exposure to Cd may cause kidney diseases and also functional changes of the nervous system, affecting mainly children and infants. To date, the mechanisms underlying its neurotoxicity are completely unclear [76]. Cd causes oxidative stress that affects the CNS and it can impair the neurotransmitter systems and neurogenesis. It has been associated to olfactory and motor dysfunction, neurological deficits, including memory loss and metal retardation, and it has also been identified as a possible factor that is involved in the etiology of several neurodegenerative diseases, including AD and PD [77]. Cd toxicity has been extensively studied in many aquatic invertebrates [78,79], but minor focus has been attributed to its neurotoxic effects. Studies in this regard have been principally conducted in organisms that belong to the Mollusca subphylum, examining the activity of cholinesterases (ChE and AChE) by in vitro or in vivo toxicity assays or investigating the possible correlation to environmental metal pollutants, including cadmium and lead, by field studies.

The bivalves of the species *Pecten jacobaeus* were subjected to in vitro exposure of Cd together with Zn (10^−9^–10^−3^ M) for 15 min. Cd exposure coincided with the dose-dependent inhibition of ChE activity in digestive gland, gills and adductor muscles, whereas less sensitivity to Zn was found [80]. Similarly, the in vitro exposure of blue mussel *M. galloprovincialis* to Cd (CdCl_2_, 10^−3^ M) for 40 min., resulting in a decrease of ChE activity in the whole body reaching the 50% inhibition. In the same study, a strong inhibition of ChE activity was also observed following to exposure to Zn and arsenic (10^−3^ M) [64].

A field study evaluated the metal accumulation (Cd, Cu, Fe, Pb, and Zn) in soft tissues of clam *Ruditapes decussatus* that were collected from different sites in the southern lagoon of Tunis. Except for Cu, significant differences in the metal concentration were found among clam tissues collected from the different polluted sites, in comparison with control samples. Biotransformation enzymes and biomarkers of oxidative stress and apoptosis were evaluated to analyze diverse toxic effects that are caused by metal pollution. In particular, the most impacted site showed the highest concentration of Cd, Fe, and Pb and a significant decrease of AChE activity was detected in the soft tissues of clams that are derived from this site, but only in the cold season. However, no significant correlation between the AChE activity and metal concentrations was found [81].

Additionally, in marine snails, *Cronia contracta,* collected from several sites of the Goa coast (India), the metal accumulation was related to the inhibition of AChE activity. In particular, high concentrations of different neurotoxic metals (Pb, Cd, Cu, Fe, and Mn) were found in the whole tissues of snails and data analysis, indicating that the metals Cd, Pb, and Cu are the most predominant inhibitors of AChE activity [28].

Mercury (Hg) is an element in the Earth’s crust and a potent neurotoxin. Diverse are the chemical forms of Hg to which human can be exposed, such as elementary (Hg), inorganic (Hg^2+^ and Hg^+^), or organic Hg compounds (i.e., MeHg). The Hg that is released in the aquatic environment is generally converted, by anaerobic microorganisms, to methylmercury (MeHg), which is more toxic than inorganic Hg [82]. Fishes having high levels of MeHg constitute the main route of organic Hg exposure for humans. Hg in its diverse chemical forms can affect the nervous system, causing the alteration of neurobiological processes, sensory deficits and cognitive dysfunctions [83]. Some studies have evaluated the bioaccumulation and toxicity of Hg in marine invertebrates [84], but only few of them focused on its neurotoxicity in this group of organisms.

The exposure of crab *C. maenas* to Hg^2+^ (0.09–0.74 mg/L) or Zn^2+^ (1.84–4.79 mg/L) for 96 h resulted in the significant reduction of eye AChE activity, which suggested the ability of both metals to alter the cholinergic neurotransmission in this organism. In addition, alteration in the glutathione-S-transferase (GST) and lactate dehydrogenase (LDH) activities was correlated to Zn and Hg^2+^ exposure [71].

High levels of Hg were found in the gills of clam *R. decussatus* that were collected from polluted sites of the southern lagoon of Tunis [81]. The accumulation of this metal was correlated with a strong inhibition of AChE activity, suggesting a neurotoxic effect in clams [85].

Arsenic (As) is a toxic metalloid that is ubiquitously distributed on Earth. The exposure in humans usually derives from As-contaminated drinking water and it is responsible for cancer, hypertension, cardiovascular diseases and also neurological effects, such as peripheral neuropathy, cognitive deficits, and memory impairment [83,86]. Similarly to other non-essential metals (Pb and Hg), numerous evidences suggest a correlation between As exposure and several neurological diseases, such as ASD, AD, ALS, and multiple sclerosis [77].

The marine biogeochemistry of the metalloid As is highly complex as well as the processes regulating its bioaccumulation and biotransformation in many marine organisms, as described, in detail, by Neff [87]. Similarly to fishes, several invertebrates, such as Polychaetes, Crustaceans, Bivalves, Snails, and Cephalopods, accumulate, in their tissues, the arsenobetaine, the nontoxic form of As. Only few data are present in literature taking the neurotoxic effects of As on marine animals into account.

A recent study evaluated the toxic effects in the clam *Ruditapes philippinarum* exposed for 28 days to As (Na_2_HAsO_4_ · 7H_2_O; 0.1 mg/L) and to nanoparticles Multi-Walled Carbon Nanotubes (NP, 0.1 mg/L), individually (As, NP) or as a mixture (As + NP). Albeit the total As concentration in the whole soft tissue of As-exposed clams resulted in being higher than control group, no significant differences were observed. Metabolic capacity (ETS, GLY and PROT), LPO and oxidative stress (SOD, CAT, glutathione peroxidase GPx, GST) were also evaluated, in addition to neurotoxicity (AChE). A significant inhibition of AChE activity was detected in all exposed clam groups, although the As exposure resulted less toxic than that to NP (single or in combination). The exposure to mixture of As + NP caused the greatest toxicity in clams, which was characterized not only by greater neurotoxicity, but also by a greater increase of oxidative stress and alteration in the metabolic activity [88].

Nickel (Ni) is a metal that is widely distributed throughout the environment and its presence in the marine ecosystems is often related to industrial pollution. Although the data on the essentiality of Ni in aquatic biota are conflicting, their toxicity is well known [89]. Elevated concentrations of Ni have harmful effects on human health, thus causing hepatotoxicity, teratogenicity, immunotoxicity, and neurotoxicity [89]. The nervous system is the main target organ of Ni and numerous are the neurological signs that are related to Ni-accumulation, such as giddiness, lethargy, and ataxia. However, only recently the effects of Ni on marine biota have been subjected to intensive studies.

The in vivo exposure of the blue mussel (*M. galloprovincialis*) to Ni (135 and 770 μg/L) for different exposure times (24–96 h and 8 days) resulted in its time-dependent accumulation in digestive gland tissue, with the Ni-uptake constant being higher in animals that were exposed to the lowest concentration. In the same tissue, the MT accumulation was also observed in a dose- and time-dependent manner. Oxidative stress was involved in Ni-exposure, as demonstrated by the significant increase of antioxidant enzymes activities (GST and CAT) and lipid peroxidation products (malondialdehyde, MDA).

Two different trends of the AChE activity were observed, depending on the Ni concentration used. A continuous inhibition of AChE activity was detected at the highest dose of Ni, while the lowest concentration only caused the increase at 48 h of exposure, followed by a decrease up to control values [90].

Among the marine crustaceans, the copepod *Trigriopus japonicus* was exposed to Ni (0.125–3.0 mg/L) for different exposure times (one, four, seven, and 12 days). Significant alterations were detected in several biomarkers of oxidative stress (SOD, GPx, GST, reduced glutathione GSH, and the ratio of reduced to oxidized glutathione GSH/GSSG), LPO levels, and MT synthesis. The Ni neurotoxicity was also evaluated and a significant increase of AChE activity was observed in the late exposure time (day 7 and 12) [91].

## 4. Marine Pollutants: Pesticides

Pesticides are chemical compounds that are specifically introduced into the environment to prevent, destroy, or control harmful organisms (pests) mainly in agriculture, including insects, rodents, fungi, and unwanted plants, as well as to block vector borne diseases in public health. They are also used for the protection of plants or plant products during their production, storage, and transport.

Although their application serves many important purposes, the very nature of pesticides is to damage unwanted organisms, posing great concern for a potential risk, even for non-target organisms. Indeed, many molecular targets of pesticides are shared between pests and non-target species, including humans, which may lead to a variety of adverse health effects. The over-use and/or misuse of pesticides is playing a negative role to the environmental health, affecting many aquatic and terrestrial species, as much to be now considered as contaminants of emerging concern (CECs), i.e., “new” pollutants in aquatic environments. Besides, the replacement of marine ingredients with plant material in the feeds used in fish farms introduces new unwanted substances in the marine environment, such as pesticides that are used in terrestrial agriculture, thus affecting not only fishes, but also other marine organisms [92]. Actually, based on all of these considerations, it is really very difficult to find the right balance between benefits and harms deriving from the use of pesticides.

Pesticides are divided into many subclasses, which are classified by the target organism (herbicides, insecticides, fungicides, and rodenticides) as well as by their chemical structure, physical state, and source of origin (i.e., botanicals). For the purpose of this review, we will focus on some classes of pesticides that are known for their neurotoxicity to marine invertebrate organisms, as, for example, organochlorines, organophosphates, and carbamates [93]. Organochlorines are synthetic pesticides, which belong to the group of chlorinated hydrocarbon derivatives, widely used in agriculture and chemical industry. Nowadays, many of the organochlorines have been banned in developed countries, as they are highly toxic and their characteristically long lifespan increases the risk of their accumulation in the environment [94]. Organophosphates and carbamates are organic molecules, with the first containing one or more phosphate ester groups, whereas the second is derived from carbamic acid. They are among the most widely used pesticides, which have been considered to be a less damaging alternative to organochlorines (together with carbamates), although they are known for their neurotoxicity, mainly targeting the cholinergic system. In particular, organophosphates and carbamate affect AChE activity by irreversibly binding the enzyme active site [95].

### 4.1. Echinoderms

Among echinoderms, sea urchin embryos and larvae are highly accounted cost-effective models for the high-throughput screening of environmental toxicants and other developmental disruptors. They are successfully used to study the mechanisms underlying the effects of diverse compounds and identify the critical embryonic stages of vulnerability (Appendix A).

Among the organophosphates, monocrotophos (MCP) is a relatively cheap insecticide that is mainly used in agriculture and animal husbandry. MCP is a water soluble pesticide and, although it is usually applied directly on soil, can also affect the aquatic flora and fauna. Its detrimental effects have been studied in numerous non-target organisms, including marine animals. The effects of different concentrations (0.01, 0.10, and 1.00 mg/L) of MCP were studied in the sea urchin *Hemicentrotus pulcherrimus*, evaluating the ACh metabolism, by determining ChAT and AChE distribution throughout development, and their activities during MCP exposure [96]. In addition, the authors investigated the MCP effects on the expression levels or activity of molecules that are involved in dopaminergic metabolic pathway. The results showed that, from the blastula to the two-armed pluteus stages, MCP causes a ubiquitous disruption of all the neurotransmitter pathways analyzed, leading to adverse effects on larval morphogenesis [96]. In the same species, MCP was shown to affect serotonergic nervous system, by inhibiting the serotonergic axon growth, reducing the number of serotonergic cells at the apical ganglion, thereby perturbing the formation of the serotonin receptor cell network. These MCP inhibitory effects lead to a consequent decrease in swimming activity of sea urchin larvae [97]. An in-depth analysis on the mechanism underlying the MCP effects on the serotonergic system revealed that the expression of netrin, a chemotropic axon guidance cue, and its receptor neogenin, were inhibited in MCP-treated embryos, which resulted in a MCP-dependent disruption of the serotonergic axon branching and synapse formation [98].

Chlorpyrifos (O,O-diethyl O-3,5,6-trichloro-2-pyridyl phosphorothioate) is one of the most used among the organophosphates due to its stability and persistence. Chlorpyrifos effects were studied on different sea urchin species (*S. purpuratus, S. droebachiensis, S. granularis, Lytechinus variegatus,* and *Dendraster excentricus*), in embryos at different developmental stages (from early cleavage to prism) that are exposed to increasing concentrations (1–160 µM) [99]. Chlorpyrifos did not affect early cleavage stages, even at the highest doses, whereas severe changes in cell phenotype and overall larval structure, i.e., the appearance of unknown pigmented cells and their extrusion from the animal pole were produced by micromolar doses [99]. In the Mediterranean *Paracentrotus lividus* sea urchin, the exposure of gastrula embryos to chlorpyrifos (from 10^−4^ to 10^−8^ M) caused dose-dependent damages, from embryos death (10^−4^ M) to the absence of coelomic cavity and archenteron invagination (10^−5^ and 10^−6^ M). Interestingly, some of the 10^−5^ and 10^−6^ M chlorpyrifos-treated larvae showed shorter than normal perioral arms. According to the authors, this malformation is a sign of nerve damage, as the nerves synchronizing the larval ciliary movements run right down perioral arms [100].

The biological effects of other organophosphates (Basudin and Diazinon) and carbamates (Carbaryl and Pirimicarb) were investigated at various levels in *P. lividus* sea urchin. In addition to affecting various aspects of embryonic development (slowing the mitotic cycle down, affecting nuclear and cytoskeletal status and DNA synthesis), Basudin and Diazinon strongly inhibited AChE activity (10^−7^ M) and induced larval defects (10^−4^ M), while carbamates showed almost no effect [101].

### 4.2. Molluscs

Among the organophosphates, the 2,2-dichlorovinyl dimethyl phosphate, which is commonly known as dichlorvos (DDVP), is a highly volatile pesticide also used in the marine fish farms/fish aquaculture, whose neurotoxic effects have been investigated in numerous organisms, including non-target invertebrates. In this regard, several studies focused on the toxic effects of DDVP exposure in commercial molluscs, at various concentrations and for different time span. (Appendix A).

In particular, the exposure of Manila clam (*R. philippinurum*) and Pacific oyster (*Crassostrea gigas*) to DDVP (0.1 and 1 mg/L) for 6 h, followed by 42 h of depuration, resulted in the paralysis of adductor muscles in both organisms. The inhibition of AChE activity was observed in the total clam tissues and oyster gills (70 and 87%, respectively). The levels of AChE were partially restored in clams after the depuration phase, but not in oysters, thus suggesting a greater sensitivity of oyster *C. gigas* to this pesticide than the clam *R. philippinurum* [102]. AChE activity reduction in the gill, but not in the mantle tissues, was also observed when *C. gigas* was exposed to DDVP (0.1–200 µM) for 96 h, as well as to other pesticides (carbofuran and oxamyl, carbamate; lindane, organochlorine), with the highest toxic effect from DDVP (inhibition rate of 88%) [103]. Two days of exposure of *R. decussatus* to 0.05 mg/L DDVP were sufficient for inducing oxidative stress, as suggested by the exponential increase in H_2_O_2_ levels in gills and digestive gland, as well as to reduce AChE activity in both tissues, which lasted until seven days of exposure. The greatest toxic effects of DDVP were observed at 0.25 mg/L, which caused significant changes of the several biochemical markers analyzed. This concentration was then selected by authors to test the protective effects of *Polygonum equisetiforme* extracts against the toxic effects of DDVP [104]. Even the exposure of mussel (*Mytilus edulis*) to DDVP (concentration gradient of 1–10,000 μg/L) for 24 h resulted in reduced gill AChE activity in a dose-dependent manner, while a persistent inhibition of AChE activity was caused by repeated exposures to DDVP. Furthermore, the DDVP exposure at the highest concentrations coincided with increased mortality, which appeared to be closely related to alteration in the closure of the shell already observed at lower doses [105].

Two forms of cholinesterases were isolated and characterized in the gill tissue of oyster *C. gigas*. Besides the differences in molecular weight, cellular localization, hydrophobicity, and state of glycosylation, these enzymes displayed different sensitivity to organophosphate or carbamate pesticides, including diisopropyl fluorophosphate (DFP), paraoxon, eserine, carbofuran, and carbaryl [106].

Azamethiphos is another organophosphate pesticide widely used in fish farms, whose toxic effects were evaluated in *M. edulis*, exposed for 1 and 24 h at 0.1 mg/L. The short-term exposure caused cytotoxicity of haemocytes, with a reduction in their phagocytic activity, thus altering the immune system, whereas no effects on their feeding behavior were observed. Again, a neurotoxic effect was associated to the decreased AChE activity in haemolymph and gill tissues in in vitro and in vivo experiments [107].

Albeit the in vitro and in vivo exposures to organophosphate (paraoxon DFP) and carbamate were assessed in *M. edulis* using combined esterase activities as biomarkers (AChE and carboxylesterases) [108,109] it was found necessary to combine haemolymph AChE activity with other biomarkers to measure exposure to organophosphates. In particular, the exposure to chlorfenvinphos (0.003–0.03 mg/L) for 24, 48, and 96 h resulted in a dose-dependent inhibition of cellular viability and immunotoxic effects, including a reduction in the hematic population, spontaneous cytotoxic activity, and alteration in the phagocytic activity. On the contrary, haemolymph AChE activity resulted in being unresponsive to the in vivo exposure to chlorfenvinphos, and a decrease was only observed at high doses (0.4–30 mg/L). Visual signs of neurotoxicity were observed at the highest exposure concentration, i.e., alterations in the mussel neuromuscular function, including the absence of adhesion to the substrate and constant shell opening [109].

The exposure of cholinergic neurons that were isolated from *Aplysia*, a marine gastropod, to paraoxon (5 mM) for 36 h resulted in morphological alterations, including cell membrane blebbing and reduced neurite growth, leading to neuronal death. Additional inhibitory effects of paraoxon were observed in isolated neurons treated with haemolymph AChE, known to positively influence neurite growth [110]. The electrophysiological analysis of R2 neurons isolated from abdominal ganglia of *Aplysia californica* permitted identifying the inhibitory effects of paraoxon on the chloride currents normally activated in response to ACh and GABA neurotransmitters [111].

Methyl parathion (O,O-dimethyl O-4-nitro-phenylphosphorothioate, MP) is a highly toxic organophosphate pesticide, which is widely applied in agriculture. The exposure of *A. juliana* to 1 and 2 mg/L of MP for 7 and 14 days resulted in its presence in the pleural–pedal ganglia [112]. Moreover, MP exposure coincided with reduced activity of AChE and increased activities of the antioxidant enzymes (superoxide dismutase SOD and CAT), in both right and left ganglia. Through proteomic analysis, several proteins, which are involved in different biological functions, appeared to be differentially expressed in both left and right pleural-pedal ganglia in all experimental conditions. The results of this study suggested different mechanisms that are involved in MP toxicity between the two ganglia, as well as different responses over time, i.e., proteins induced at seven-day MP exposure were related to damage and stress, while, at 14-day, they were related to cellular repair [112].

Besides the organophosphate pesticides presented so far, the studies focused on other classes of pesticides and their neurotoxic effects in marine invertebrates are very few, or even scarce. The first study to test the organochlorine pesticide lindane (1.0–2.5 µM) on *C. gigas* for 12 days, reported a slight decreased AChE activity in the gills (about 30%) [103]. The herbicide atrazine (ATZ) toxic effects were investigated in the green mussels (*Perna viridis*), alone or in combination with other emerging contaminants, including the plasticizer bisphenol A (BPA) and the drug carbamazepine (CBZ). In particular, the exposure to ATZ for seven days resulted in significant alterations of DNA damage biomarkers, detoxification enzyme Ethoxyresorufin-O-Deethylase (EROD), and a strong inhibition of haemolymph AChE activity at environmentally relevant doses or below. Only the exposure to high doses of ATZ was associated to alterations in mussel immune functions, including the increase of the total number of haemocytes and a decrease in their phagocytic activity [113].

### 4.3. Crustaceans

The neurotoxic effects of DDVP were also investigated in invertebrate organisms belonging to the crustacean subphylum. (Appendix A) In particular, the lobster larvae (*Homarus gammarus*, stage V) that was exposed to DDVP (0.01–100 μg/L) for 24 h showed the inhibition of AChE activity starting from 10 μg/L, followed a by a dose-dependent trend, with 50% inhibition at 2.7 μg/L [114]. Additionally, short-term exposure (6 h) to DDVP (0.1–50 μg/L) caused a significant reduction (about 40%) of AChE activity at 50 μg/L in the larval stage IV. Moreover, the significant inhibition of AChE activity, but no mortalities, was observed in lobster larvae following repeated exposure to 50 μg/L of DDVP [115]. Additionally, the exposure of adult amphipods (*Hyale nilssoni*) to DDVP (5–320 μg/L) for 96 h resulted in a dose-dependent reduction of AChE activity [116].

The characterization of a soluble form of ChE in the eye of the white shrimp (*Litopenaeus vannamei*) suggested the ocular tissue as an additional and ideal tissue for evaluating the neurotoxic effects of substances in invertebrate organisms [117] as well as the ChE enzyme as a reliable biomarker of neurotoxicity. In fact, the exposure of *L. vannamei* to the OG methamidophos (0.66–1.35 mg/L) for 24 h resulted in the dose-dependent inhibition of eye ChE activity, with the maximum inhibition of 48.7% at the dose of 1.18 mg/L. Significant changes in behavior, including altered swimming movements, hyperactivity, and spasms, were correlated to pesticide exposure, while mortality and feeding rate were not susceptible endpoints at the conditions tested in *L. vannamei* [117]. The ChE enzyme, also identified and characterized in the prawn *Palaemon serratus* eye, was responsive to the exposure to various organophosphate (cholrpyrifos-oxon, malaoxon, triazophos-oxon, DDVP, paraoxone methyl, and paraoxone ethyl) and carbamate (serine sulphate, carbofuran, propoxur, and carbaryl) pesticides, with a greater susceptibility toward chlorpyrifos-oxon and carbofuran [118]. Furthermore, the exposure of *P. serratus* to DDVP (0.057–0.452 μM) for 24 h resulted in the inhibition of ChE activity, which was in close correlation to prawn mortality. Moreover, the combined exposure of DDVP with mercury (1–37.5 μM) did not cause synergistic effects in the inhibition of eye ChE activity, which suggested independent toxic mechanisms of the two contaminants [119].

The toxic effects of fenitrothion (O,O-dimethyl O-4-nitro-methyl phosphorothioate) were evaluated by exposing *P. serratus* to sublethal concentrations (39–625 ng/L), which resulted in significant alterations in the swimming velocity and inhibition of AChE activity in both eyes and muscle. The exposure of prawns to fenitrothion showed impairments in the avoidance behavior and alterations in energy metabolism (LDH and isocitrate dehydrogenases, IDH) [120].

The effects of chronic exposure to azamethiphos at low concentrations on lobster populations living nearby caged salmon that was treated with pesticides were a major concern. Thus, adult lobsters (*Homarus americanus*) were exposed to sub-lethal concentrations of azamethiphos (61 ng/L) for 10 days, resulting in an increased rate of mortality, associated with significant changes in biomarkers of stress/hypoxia (serum total proteins, haemocyanin, and lactate) and oxidative damage (protein carbonyls in gills and serum) [121]. Azamethiphos-treated lobsters showed a significant inhibition of muscle ChE activity (50%), alterations in the hepatosomatic and gonadosomatic indices and hepatopancreas lipid levels, whereas no morphometric alterations were identified. Even after 24 h of depuration in clean seawater, all of the sub-lethal effects associated to the azamethiphos chronic exposure of lobsters remained unchanged, thus increasing the risk of cumulative/additional effects of other stress sources [121]. Two forms of AChE enzymes were isolated and characterized in the sea lice (*Lepeophtheirus salmonis*), showing diverse sensitivity to azamethiphos, thus providing a further understanding regarding a resistance mechanism to organophosphate pesticides in these common parasites [122].

The toxic effects of Chlorpyrifos were evaluated in different crustaceans. The invasive *Artemisia franciscana* and autochthonous *A. parthenogenetica* were exposed to chlorpyrifos (0.1, 1, and 5 µg/L) to shed light on a possible mechanism of competition between them [123]. Both of the species tolerated high concentrations of pesticide, but the survival rates and fecundity were higher in the *A. franciscana*. Moreover, the inhibition of AChE activity was dependent on sex, stage of development, and concentration of the pesticide, and they reached 80–85% in adult forms of both species exposed to the highest concentration of chlorpyrifos [123]. The exposure of adult shrimps (*Litopenaeus vannamei*) to sub-lethal concentration of chlorpyrifos (0.7 and 1.3 μg/L) for four days caused significant alterations in the LPO levels and antioxidant enzymes activities, such as CAT, GPx, and GST, analyzed in diverse tissues: muscle, hepatopancreas, and gills. Furthermore, a significant inhibition of AChE activity in the brain was observed [124].

The *Artemia salina* at the cyst stage was exposed to several pesticides, including chlorpyrifos oxon, diazinon, and carbaryl, at doses similar to those found in contaminated coastal environment (10^−11^–10^−5^ M), for long exposure times (72, 96, and 192 h). Significant changes in the hatching speed and larval survival were observed, as well as a decreased ChE activity at 96 h in the protocerebrum, a brain segment that is associated with vision [125]. The copepod *Tigriopus japonicas* was exposed to diverse pesticides (alachlor, chlorpyrifos, dimethoate, endosulfan, lindane, and molinate) for 24 h at different concentrations. Except for dimethoate and molinate, the highest concentrations of pesticides caused the significant inhibition of AChE activity [126].

Among the pyrethroid pesticides, the exposure of common prawn (*P. serratus*) to deltamethrin (0.6–313 ng/L) for 96 h caused a significant reduction in the swimming velocity that was correlated to antioxidant enzymes activities (GST, GPx, and CAT) and energy metabolism (IDH and LDH). No correlation was found between this behavioral parameter and neurotoxicity; at different concentrations of deltamethrin, eye AChE and muscle ChE activity were significantly increased [127]. The effects of exposure to deltamethrin (0.1 µg/L for four days) were also analyzed in black tiger shrimp (*Penaeus monodon*) in combination with abiotic factors, such as temperature and salinity. The exposure of pesticide resulted in the alteration of some oxidative stress biomarkers (GSH, CAT, GPX) and in the significant reduction of the muscle AChE activity that resulted in also being influenced by abiotic factors [128].

Although the great amount of data on pesticides effects on marine invertebrate organisms, mainly at relatively short-term exposures (from few hours to 14 days), much remains to be comprehended regarding the potential effects of lower-level exposures, but at much longer-term.

## 5. Marine Pollutants: Microplastics and Nanoplastics

The widespread human use of plastics has led to the appearance of a new ECs, the microplastics (MPs). These particles of small dimensions (<5 mm) principally derive from large plastics that are reduced to small sizes up to the nano order (nanoplastics, <100 nm) through hydrolysis, mechanical forces, or ultraviolet radiation. Polyethylene (PE), polypropylene (PP), and polystyrene (PS) are among the most prevalent plastics. The scientific community highlights that the plastic particles can alter terrestrial ecosystems and, in particular, the marine environment, affecting organisms, including zooplankton, molluscs, and fishes [129]. Numerous adverse effects of MPs have been demonstrated in the marine biota, including nanoparticle (NP) accumulation, oxidative stress, inflammation, gene damage, and neurotoxicity, with possible consequences on human health [130,131,132]. Furthermore, physical and chemical properties of MPs give them the ability to absorb and carry other contaminants, such as heavy metals, pesticides, PPCPs, and entire microorganisms, thus increasing their toxic potential [133] (Appendix A).

Avio et al. [134] conducted the first study that aimed to investigate the neurotoxic effects of MPs in the molluscs, in which Mediterranean mussels (*M. galloprovincialis*) were subjected for seven days to exposure of PE and PS MPs (100 μm, 1.5 g/L), free or in association with pyrene. According to MPs as carriers of marine pollutants, the ability of both type of MPs (PE and PS) was demonstrated to accumulate the pyrene in time- and concentration-dependent manner. The mussel digestive gland was identified as the principal organ of MP accumulation, even if gills and haemolymph also presented a small number of particles. All of the MP treatments (both virgin and contaminated) caused numerous biological effects in the Mediterranean mussels, including DNA damage, moderate change in the oxidative state, and alteration of lysosomal compartment. The greater genotoxic response was only observed after exposure with pyrene-treated polymers in comparison to free MP treatment. In support of the biochemical and cellular analyses, alteration in gene expression that are associated with different pathways (lysosomal metabolism, immunological functions, genotoxicity, antioxidant, and detoxification systems) were obtained in the MP-treated mussel groups. The neurotoxic effect of both PE and PS treatments was suggested by the decrease in the AChE activity in the gill tissues without any effect being produced by the pyrene contaminant. No changes in the AChE activity were detected in the haemolymph in all experimental treatments [134].

Alterations in gene expression were also found in the same species *M. galloprovincialis*, exposed for a short period of time (96 h) to different concentrations of PS (0.11 μm, 0.05–50 mg/L), free or in association/combination to carbamazepine (Cbz, 6.3 μg/L), an anticonvulsant drug. In particular, the MP exposure affected the transcriptional levels of genes that are associated to biotransformation and innate immunity in the gill tissue (cyp11, cyp32, cat, and lys) and to cell stress-response in digestive gland (HSP70), but not in a dose-dependent manner. The treatment with PS increased the oxidative stress and LPO in the digestive gland, as suggested by biochemical analyses. The decreased ChE activity was only revealed in the PS-treated haemolymph, suggesting the ability to the particles to alter the cholinergic signaling at low concentrations (0.05 and 0.5 mg/L PS). No evidences were reported around the presences of microparticles in the mussel tissues [135].

Bivalves of the species *Scrobicularia plana* were subjected to the exposure of PS (20 μm, 1 mg/L) for 14 days, followed by seven days of depuration (for a total of 21 days). These MP particles were detected in clam tissues, including haemolymph, gills, and digestive gland by different experimental analyses. The gills were suggested to be the primary clam organ of polymer accumulation as well as being more effective against the oxidative stress than the digestive gland. The activities of antioxidant enzymes in the gill tissues (SOD, CAT, GPx, and GST) increased, probably contributing effectively to reduce/limit oxidative stress. In this tissue, the decrease in the AChE activity was detected at three and 14 days of PS exposure and after the depuration after 21 days. In the digestive gland, an increase of the LPO levels was observed at seven days and the lower levels of all enzymatic activities except of the GPx activity [136].

A recent study investigated, in *S. plana*, the effects of 14-day exposure to low-density polyethylene (LDPE) MPs (11–13 μm, 1 mg/L), free and in association with oxybenzone (BP-3), a compound that is utilized as Sunscreen UV Filter. The activities of antioxidant and biotransformation enzymes, together with those of oxidative stress and DNA damage repair, were assessed in clam tissues, including digestive glands, gills and haemolymph. At seven days of exposure to LDPE associated with BP-3, the increase in the LPO and antioxidative systems (SOD and GPx) was observed in comparison to the LDPE virgin and control groups. The gill tissue showed a greater sensitivity to MP exposure (both virgin and BP-3 absorbed) when compared to the digestive gland, as suggested by the analysis of all the different tested biomarkers. Unlike other studies regarding the neurotoxicity of MPs in molluscs, no inhibition in the AChE activity was revealed in the gills under these experimental conditions, probably due to LDPE size; an increase in AChE activity was observed only at 14 days of exposure to BPE-contaminated MP [137].

Because of the great variability obtained by the different quantification methods of MPs, Tlili et al. [138] compared three different protocols of isolation and enumeration of NPs in the wedge clams (*Donax trunculus*), exposed for 15 days to MP mixture of PP and PE (PP/PE mixture, ratio 1:1) (100–400 mm, 0.06 g/Kg of sand). Their data suggested that the digestion method with H_2_O_2_ was more efficient for estimating the MP accumulation in the different clam tissues (gills, digestive gland, and flesh) than the direct observation or than the acid mixture (HNO_3_/HCl) digestion method. From the biological point of view, it was demonstrated that the primary site of MP accumulation is the gill, whereas no particles were found in flesh tissue. The exposure to PP/PE mixture is responsible for the increase of the oxidative stress biomarkers, such as CAT and GST, in all clam tissues examined, even if at different time of MP exposure. A significant neurotoxicity of MP mixture was revealed by the reduction of AChE activity in both gills and digestive gland; no change was observed in the flesh tissues [138].

Using the fluorescent labelled NPs, the internalization of polystyrene-NPs (PS-NPs, 80-nm) has been evaluated in different organs of the *C. fluminea* exposed at different concentrations (0.1, 1, and 5 mg/L) for 96 h. By using fluorescence imaging and histological analysis, it was shown that visceral mass (intestine and stomach) and gills were the main organs of accumulation of PS-NPs, whereas NPs were also observed in the mantle only in traces. These results were correlated to several biomarkers of oxidative stress (SOD, CAT, GSH, GPx, glutathione reductase GR, and GST) and MDA, indicators of liver damage (aspartate aminotransferase, GOT, and alanine aminotransferase, GPT) and intestinal inflammation (diamine oxidase, DAO, and D Lactate). The neurotoxicity of PS-NPs was evaluated and a significant inhibition of AChE activity was only detected only in the visceral mass and mantle, but not in the gills [139].

Neurotoxicity in association to immunotoxicity was investigated in bivalve species, *Tegillarca granosa* exposed to PS-MP microparticles (490 ± 11 nm, 1 mg/L), virgin, and in association with BPA (10 and 100 ng/L) for 14 days. MP and BPA accumulation was detected in the soft tissue of clams after 14 days of treatment and changes in the hematic parameters were analyzed, including a reduction in the total amount of hematic population, alteration in their cell-type composition, and phagocytic activity. In addition, genes that belong to the immune system-related NF-kB signaling pathway, such as TNF receptor-associated factor 6 (TRAF6), TGF-Beta Activated Kinase 1 (MAP3K7), Binding Protein 2 (TAB2), and IkappaB Kinase (IKKa), including NF-kB, were all significantly suppressed by BPA and MPs, individually as well as in combination. The diverse treatments also affected the clams NS: BPA and MPs unaffected the expression of γ-aminobutyric acid transaminase (GABAT), the modulatory enzyme of the neurotransmitter GABA, but their co-exposure significantly downregulated it. Similarly, the expression level of AChE was only inhibited by a high dose of BPA and by the co-exposure of BPA and MPs [140].

Other marine organisms, belonging to the Crustacea phylum, have been used to test the neurotoxicity of MPs. In particular, larvae of striped barnacles (*Amphibalanu ampitrite*) and brine shrimp (*A. fransiscana*) were exposed to various concentrations (0.001–10 mg/L) of fluorescent PS MPs (0.1 μm) for 24 and 48 h, which resulted in the accumulation of MP in the intestine of both crustaceans without compromising their survival. Significant alterations in the swimming behaviour were detected in both planktonic larvae after exposure to high MP concentrations. The additional toxic effects of MPs were analyzed through changes in the activity of oxidative stress and neurotoxicity biomarkers, respectively CAT and AChE and pseudocholinesterase (PChE). In particular, a significant increase of CAT activity was detected at almost all of the concentrations tested and in both crustaceans, thus suggesting the ability of MPs to cause oxidative stress. Although a significant change in cholinesterase activities (AChE and PChE) was observed after exposure to specific MP concentration in *A. amphitrite*, neurotoxic effects were not with the same trend or in a dose-dependent manner [141]. In *A. amphitrite*, an increase of both enzymatic activities was observed at low concentration of MPs, and a decrease only at high doses. In *A. franciscana*, the enzymatic activity of cholinesterase does not have a similar tendency, since a significant reduction of AChE activity was only observed at low doses, whereas the PChE activity was mainly not different from that of control group (increase only at 0.01 and 0.1 mg/L) [141].

Among the marine planktonic crustaceans, *A. franciscana* was also subjected to short- and long-time (48 h or 14 days) exposure of cationic amino-modified PS NPs (PS-NH2, 50 nm) at different concentrations (0.1–10 μg/mL). No experimental evidences of intracellular accumulation of NPs were observed. After short-term exposure to PS-NH2, a significant decrease in the naupliar growth and morphological changes in the development were detected in a dose-dependent manner. The rate of mortality resulted in being increased in long-term exposure, whereas no alteration in the growth and feeding behaviors were found. Oxidative stress was involved in short-term PS-NH2 exposure, as demonstrated by a reduction in the antioxidant enzyme activity, whereas an evident increase of lipid peroxidation (thiobarbituric acid reactive substances, TBARs) was observed in long-term exposure. The alteration of AChE activity suggested possible neurotoxic effects after short- and long-time exposure without a dose-dependent trend. Finally, all of these time-dependent changes in physiological processes and biochemical markers were correlated to the expression of genes that are related with stress response (HSP26, HSP70), molting (cstb), and development (clap, tcp) that are caused after PS-NH2 exposure [142].

The studies cited so far increasingly show how critical the situation of the marine environment is, how many knowledge gaps still exist to be filled by scientific research, and how important it is to implement new technologies to address the risk assessment of new ECs.

## 6. Omics Studies

Genomics, transcriptomics, proteomics, metabolomics, lipidomics, epigenomics, and phenomics are multiple omics analyses that, if integrated, can lead to a comprehensive understanding of the crucial mechanisms that underlie the response of organisms to contaminant exposure. Nevertheless, few studies in the ecotoxicological field have been performed using omics analyses in marine invertebrates. When applied, as they provide biological responses at multiple levels, such analyses have also led to new evidence of alterations in nervous system homeostasis that is induced by the examined contaminant, in addition to the identification of the molecular pathways involved. Here, we focused on the omics studies showing neurotoxic implications of pollutants in the marine invertebrate organisms that are covered by this review, such as molluscs, crustaceans, and echinoderms.

A transcriptome analysis was conducted in the oyster *Crassostrea angulata* to investigate the biological mechanisms underlying the bioaccumulation of Cu. Different organisms of oyster were exposed or not to Cu (30 μg/L for 30 days) and two groups accumulating different levels of Cu (high or low) were selected. Gene ontology (GO) analysis of the transcriptomes was performed on gills and mantle tissues and it revealed that the functional categories of the high-Cu enrichment mantles genes were related to lipid and neurotransmitter transport, vesicle, response to chemical stimulus, and ATPase activity. Among the neurotransmitter transport group, genes encoding the proline transporter and GABA transporter 2 (GAT2) were also identified. In the mantle, the expression of GAT2 was significantly increased in high-Cu accumulating individuals as compared to the low-Cu accumulating ones. The experiments of gene silencing of *GAT2* showed the significant reduction of Cu levels in gill and mantle, suggesting a role for Cu in the activity of the GABA neurotransmitter [143].

Another Cu bioaccumulation study was conducted in oysters (*C. hongkongensi*) that were exposed to Cu (10 and 50 µg/L) for four weeks. It is known that the Cu accumulation causes a disturbance in lipid metabolism [144]. In marine environment, lipids are often considered to be an indicator of nutritional status and ecological health [145], and only recently they are considered biomarkers of pollutant contamination. In vertebrates, lipids are the main constituents of myelin and, in several studies, the perturbation of membrane lipid composition (cellular and BBB) and changes in lipid metabolism have been involved in the disruption of myelin homeostasis [146]. In this study, different oyster tissues were analyzed for the metal content and lipid types. The greatest accumulation of metals was found in the mantle, followed by gills, adductor muscles, and gonads; whereas, the digestive gland had a relatively low percentage of total tissue Cu. Oyster digestive gland exposed to 50 µg/L Cu showed a reduced lipid content with respect to 10 µg/L Cu exposed glands. On the contrary, the exposition to 50 μg/L Cu did not affect LPO in oyster with respect to the control; whereas, after exposition to 10 μg/L Cu, lipid was slightly induced in week 2, but decreased in week 4. Lipidomic analysis was performed on digestive gland of oyster that was exposed to 50 μg/L Cu for one or four weeks. The analysis showed 142 differentially altered lipid species after one week of Cu exposure; 46 of those were increased, whereas 96 species were reduced. Upon four weeks of Cu exposure, there was a much weaker response [147]. The lipid species that are affected by Cu exposure included those belonging to subcellular compartmentalization, the modulation of inflammatory responses, and phospholipid remodeling.

Omics studies have been used to also study the toxic effects of marine contaminants in invertebrate organisms that belong to the Crustacean phyla. The sublethal exposure of the copepod *Calanus finmarchicus* to Hg^2+^ (HgCl_2,_ 5 μg/L) for 48 h resulted in transcriptional changes, as shown by microarray (oligoarray) and quantitative RT-PCR. Data analyses allowed for the identification of diverse pathways that are involved in the toxicity caused by the Hg exposure, including ATP production, oxidative stress, apoptosis and autophagocytosis, glutamate toxicity, calcium homeostasis, and cell signaling. In particular, in Hg-exposed copepods, it was observed the upregulation of two genes involved in the transport of glutamate from the synaptic cleft: high-affinity glutamate transporter (*EEAT3*) and neurotransmitter symporter (*CPIJ015063-PA*). The up-regulation of these genes was probably induced to remove the excess neurotransmitter from the synaptic cleft, thus supporting the hypothesis that Hg causes the glutamate-mediated neurotoxicity, as yet observed in mammals and vertebrates [148]. Moreover, in agreement with the hypothesis that Hg increases the release of neurotransmitters, in Hg-exposed *C. finmarchicus,* it was observed the downregulation of *HSP70*, which is known to be involved in neurotransmitter secretion in vertebrates [148].

Some omics studies were also conducted in Echinoderms. A metabolomics study was performed in the embryos of the black sea urchin *Arbacia lixula* exposed to different concentrations of CuO NPs (0.7–20 ppb) for 72 h [149]. Alterations in the neurotransmission and skeletogenesis of embryos were detected. In particular, the nuclear magnetic resonance (NMR)-based metabolomics analysis allowed for revealing a significant dose-dependent decrease in choline and N-acetyl serotonin levels in embryos exposed to the two highest CuO NP concentrations, as compared to control embryos [149]. In agreement, the inhibition of AChE activity and the decrease of serotonin immunostaining were observed in the plutei of the same species exposed to CuO NPs by Maisano et colleagues [150].

The genomic response following UV-B exposure (20 μW cm^−2^ for 2 h) was analyzed in sea urchin *S. intermedius* [151]. It is well known that climate changes alter the marine ecosystem, by increasing ocean warming and acidification, and by enhancing UV-B penetration. The effects of UV-B on the behaviors of the marine invertebrate species in the intertidal and shallow seas have been reported [152]. In particular, authors investigated the transcriptomes of UV-B exposed sea urchin exhibiting covering (moving objects onto their dorsal surface) and sheltering (living in shelters) behaviors, as compared to the unprotected group. The analysis revealed 330 differentially expressed genes (DEG), which were classified into groups of some biological processes, including signal transduction, translation, endocrine, and nervous systems [151]. Among endocrine and nervous systems, two important genes were identified: the transient receptor potential ankirin 1 (*TRPA 1*) and the *opsin*. *TRP1* belongs to a group of transient receptor potential (TRP) ion channels that are present on the plasma membrane and involved in the detection and transduction of thermal information to the nervous system [153]. The opsin is a photoreceptive protein that acts converting a photon of light into electrochemical signal. The opsin expression was demonstrated by Ullrich-Lüter et al. [14] in the photoreceptor cells (PRCs) that were located in tube feet of *S. purpuratus* sea urchin. Zhao and co-authors [151] found that the expression of both *TRPA1* and *opsin* genes was significantly higher in sheltered sea urchins than in unprotected individuals, which suggested that their expression ensured sensitiveness of UV-exposed sea urchins.

The majority of omics studies have been performed by analyzing the effects of single species of pollutants on marine organisms (as the few examples previously described), rarely considering mixtures of contaminants. An emblematic issue in the ecotoxicological field is that, in the realistic environmental conditions, multiple pollutants act in association, causing mixed effects. Even in this case, few omics studies can be reported.

A high-throughput screening approach was used to analyze the combined effects of Cd and UV-B in the sea urchin *P. lividus*. The embryos were exposed to Cd (continuous exposure to CdCl_2_, 10^−4^ M) and UV-B radiation (exposed at 32 cell stage, i.e., about 3 h postfertilization, at the dose of 200 and 400 J/m^2^), individually or in combination (Cd/UV, i.e., 10^−4^ M Cd Cl_2_ continuous exposure + 200 J/m^2^ UV-B at 32 cell stage), and 127 selected genes were analyzed by NanoString technology. Two genes that were correlated to nervous system were specifically downregulated: the transcription factor *onecut*, which is prominently expressed in the ciliary band of pluteus embryos and in the apical organ and the elongating arms [59], and *SSPO*, which encodes a secreted glycoprotein of the extracellular matrix with a conserved role in axonal pathfinding during neuronal development in the chordate phylum [154]. In particular, the downregulation of *onecut* gene was only observed in Cd-exposed embryos, while the *SSPO* gene was downregulated in both Cd and Cd/UV-exposed embryos [155]. 

A NMR-based metabolomics on green mussels *P. viridis* exposed to Cd (20 μg/L) and Cu (50 μg/L), individually or combined (Cu + Cd), for 1–2 weeks, revealed changes in the metabolic profile of the soft tissue. Cd and Cu exposure both induced disturbances in osmoregulation, amino acid, and energetic metabolisms. A significant alteration of glutamate and glutamine levels was also revealed suggesting a neurotoxic effect of these metals in mussels. After two weeks, the Cd-exposed group showed a recovery of some of the metabolic disturbances. Moreover, the data analysis revealed the high similarity between the metabolic profiles that are induced by Cu and by Cu + Cd mixture, suggesting the dominant effect of Cu in the metabolic disturbances [156]. An alteration of glutamate and glutamine levels was also found when analyzing the metabolic profiles of the adductor muscle in the same organism that was exposed to Cd (20 μg/L) for four weeks [157].

A multifactorial analysis was applied to assess the metabolic response of mussel *M. galloprovincialis* exposed for seven days to a wastewater treatment plants (WWTP) effluent extract in laboratory conditions [158]. The WWTP was chemically characterized, identifying different organic contaminants, including drugs, chemotherapy drugs, stimulants (caffeine, nicotine), herbicides, and insecticides. The relevant metabolic pathways were significantly inhibited or disrupted, such as amino acids metabolism (phenylalanine, tyrosine, tryptophan, arginine and proline), purine and pyrimidine metabolism, citric acid cycle intermediates (malate and fumarate), and glutathione metabolism, suggesting the alteration of biological mechanisms, such as energy metabolism, oxidative stress defenses, DNA and RNA synthesis, and immuno responses. Moreover, a significant alteration of neurohormones was observed, in particular a down-modulation of dopamine (−36%) and a serotonin metabolite 5-methoxyindoleacetate (−57%) levels. Dopamine and serotonin are both neurotransmitters regulating neuromuscular functions (valves closing, siphon activity, and ciliary movement) and, therefore, this study confirmed the neuromuscular and feeding behavior alterations that were observed in molluscs following exposure of contaminants, especially as pesticides, as previously described [158].

One of the challenges in environmental omics studies is to characterize the observed disturbances at the molecular level and relate them to their different origins, in order to adequately decipher the biological responses to environmental stressors undertaken by marine organisms. When multiple pollutants come into play, an appropriate strategy, together with the experimental design as well as an accurate data analysis, are crucial in efficiently distinguishing the different effects occurring simultaneously.

## 7. Conclusions

The fate and behavior of emergent pollutants in the environment as well as the effects on the biological systems are largely under-documented and poorly understood. In addition to not being degradable, many of these pollutants tend to accumulate in the environment and, due to the biomagnification effect, they move up the food chain, progressively increasing their concentrations in the tissues of living organisms with the increase of the trophic level, also becoming toxic to the human beings. Indeed, fish and invertebrate marine organisms represent the main contaminated foods that are consumed by humans, with the risk that some of these contaminants pass, through placenta transfer or lactation, to the progeny [159,160]. It is noteworthy that many of the pollutants described in this review are involved in vertebrate myelin homeostasis disturbance, such as hypomyelination and demyelination [146]. Indeed, several neurological disorders that are associated with myelin status, including intellectual and cognitive disabilities and altered behaviors, have been associated with prenatal exposure to some contaminants, such as heavy metals and pesticides [146]. Myelin is essential for proper NS performance, with the function of facilitating the conduction of electric signals over long distances in large animals. Interestingly, myelination has been found in some very small crustaceans, as copepods [161].

The present work has only highlighted some of the marine invertebrates considered excellent model systems in the neurobiology field, which, therefore, can allow a broader spectrum of analyses on neurotoxic substances that are present in the marine environment. However, the situation is quite complex, because pollutants are not present as single substances in the environment, but they often act in association and may have synergistic/antagonistic effects or simply interact each other, and that is very difficult to characterize.

## Figures and Tables

**Figure 1 biology-10-00161-f001:**
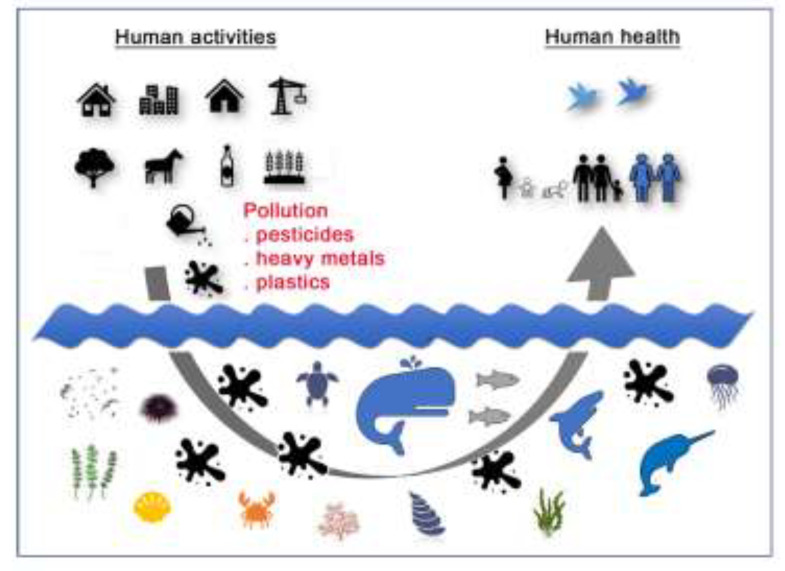
Diagram describing a wide variety of hazardous anthropogenic pollutants that pervade the entire water column, including sediments, originating from different levels of human activities releasing industrial, agricultural, hospitals, clinics and domestic wasted into the sea, with potential and adverse effects on human health.

**Table 1 biology-10-00161-t001:** Neurotoxic metals and metalloid *.

Metals	
Essential	Non-Essential
Copper	Arsenic *
Iron	Cadmium
Manganese	Lead
Zinc	Mercury
	Nickel

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
