# Peer review of "Neurotoxicity in Marine Invertebrates: An Update"

_biology, 2021, doi:10.3390/biology10020161_

Round 1

Reviewer 1 Report

This manuscript takes on too many subjects and as a result, most subjects are treated somewhat superficially. The commendable part of this manuscript is its basic review of pesticide and microplastics toxicology among representatives of major groups of marine invertebrates. The pesticides section (beginning line 541) and the microplastics section (beginning line764) are fairly well organized and presented, as though they were written by specific co-authors with experience in these topics, as well as writing experience. The metals section that begins on manuscript line 390 is much less successful. Rather than a true review, the authors set the pattern of summarizations of data in the studies they cite, with no or low insights. For example, the first toxicant considered, Cu, could be introduced by alerting a reader that the threshold hemolymph concentration of Cu regarded as external (because many invertebrates use hemocyanin as a respiratory pigment) that causes deleterious effects varies with systematic group and life stage … but there is no such preamble or summary given. Furthermore, information is not organized within a topic according to any conventional standard: by life stage, by physiological system affected, by systematic group. Without perspective and insightful organization, there is little advantage to a reader of such a metals review beyond the bibliography. In revision of the metals section, the style of the pesticide and microplastics toxicology sections should be emulated, but even in these latter sections, it is important to summarize with perspective on the main impacts of these classes of pollutants, which is not done to sufficient extent.

Lines 380-382: Here is the goal of this manuscript. The revision should focus more narrowly on this topic and introductory material that serves it. All the introductory material (lines 45-353) should be revamped to reflect the narrower focus.

It is not clear why the superficial phylogeny and systematics of larval and postlarval invertebrate groups (lines 122-272) is included. What is the goal of relating tidbits of invertebrate phyla systematics, the NS design of echinoderms, mollusks and crustaceans, plus a few details of their sensory systems, etc? I believe they are meant as introductory material to the effects of metals and other pollutants on the NS, but almost all of this is completely unnecessary, and covered much better elsewhere in dedicated articles and textbooks on these subjects, to which the authors could refer. These descriptions are not necessary to the later discussions of pollutants that act on neurotransmitters of invertebrate NS. The neurotransmitter section has merit as a lead-in to marine pollutants, but if the other introductory material were eliminated, this subject could be treated in greater and more beneficial detail.

226-8: I do not believe that Eccles, Hodgkin, or Huxley ever worked on Aplysia. H&H won the Nobel for their work in the squid giant axon. Eccles’ principal subjects were the frog NMJ and cat spinal cord in which he and his collaborators developed pioneering intracellular recording techniques. Eric Kandel, however, did receive the Nobel in 2000, largely for his work in Aplysia learning and memory. Is it possible that it is Kandel and the 2000 Nobel the authors are thinking of?

335 “outer layer” substitute a term such as crust or biosphere or change sentence to better describe what is meant

971 beginning with this paragraph, a number of topics are taken up and disposed of without much connection to preceding sections. Perhaps they could be better organized to constitute a separate section on mixed effects of multiple toxicants and of the influence of abiotic factors on the toxicology of groups of pollutants. I actually think that may have been the intention of the authors, to discuss mixtures and modifying influences, but the material needs to be organized around a cohesive theme.

The entire manuscript would benefit greatly from editing by an English language editor. Nearly every sentence up to line 540 has a problem. Most are quite minor, but cumulatively, the mistakes exact a toll on the reader.

Author Response

R1) This manuscript takes on too many subjects and as a result, most subjects are treated somewhat superficially.

  1. A) we extensively modified the manuscript, adding, removing or correcting almost all sections, trying to correct these gaps.

R1) The metals section that begins on manuscript line 390 is much less successful. Rather than a true review, the authors set the pattern of summarizations of data in the studies they cite, with no or low insights. For example, the first toxicant considered, Cu, could be introduced by alerting a reader that the threshold hemolymph concentration of Cu regarded as external (because many invertebrates use hemocyanin as a respiratory pigment) that causes deleterious effects varies with systematic group and life stage … but there is no such preamble or summary given.

  1. A) The metals section has been extensively revised, and re-organized. New insights have been added throughout the section.

R1) Furthermore, information is not organized within a topic according to any conventional standard: by life stage, by physiological system affected, by systematic group. Without perspective and insightful organization, there is little advantage to a reader of such a metals review beyond the bibliography.

  1. A) This section is organized in two sub-sections, i.e. essential and non-essential metals, and within each of them we tried to group the data according to the phyla. Information concerning functions within organisms and related neurological pathologies have been added.

R1) In revision of the metals section, the style of the pesticide and microplastics toxicology sections should be emulated, but even in these latter sections, it is important to summarize with perspective on the main impacts of these classes of pollutants, which is not done to sufficient extent.

  1. A) we added what the reviewer requested.

R1) Lines 380-382: Here is the goal of this manuscript. The revision should focus more narrowly on this topic and introductory material that serves it. All the introductory material (lines 45-353) should be revamped to reflect the narrower focus.

  1. A) all the introductory sections (preceding lines 380-382 in the first version of the manuscript) have been revised, re-organized and/or re-written according to the reviewer suggestion.

R1) It is not clear why the superficial phylogeny and systematics of larval and postlarval invertebrate groups (lines 122-272) is included. What is the goal of relating tidbits of invertebrate phyla systematics, the NS design of echinoderms, mollusks and crustaceans, plus a few details of their sensory systems, etc? I believe they are meant as introductory material to the effects of metals and other pollutants on the NS, but almost all of this is completely unnecessary, and covered much better elsewhere in dedicated articles and textbooks on these subjects, to which the authors could refer. These descriptions are not necessary to the later discussions of pollutants that act on neurotransmitters of invertebrate NS.

  1. A) In our opinion, the information included in this paragraph (renamed as “General remarks of marine invertebrates as model organisms and basic notes of their nervous system”) are important to have an immediate view of the main structures of the NS in the different marine invertebrates selected in this review. We are aware that this topic is better covered by dedicated articles and textbooks, to which of course we refer, but we think that the description of the NS is a useful introductory and preparatory part for the following paragraphs of this review. It only serves to recall some notions of the NS of the marine invertebrates described here. Indeed, this review describes the effects of pollutants on nerve regions/structures, affecting both larvae and adults of marine invertebrates (echinoderms, molluscs and crustaceans) and we therefore think it might be useful for a reader unfamiliar with these topics to have a brief description of the NS anatomy, albeit superficial, in this same review. Even the figures, albeit very stylized, are explanatory of the typical nervous structures of the aforementioned invertebrates. However, we added this explanation at the beginning of the section. Besides, we have extensively revised this section too, reducing the descriptions of the NS, eliminating those unnecessary descriptions for the main text, such as the “Neuropeptides” section, which has been completely removed.

R1) The neurotransmitter section has merit as a lead-in to marine pollutants, but if the other introductory material were eliminated, this subject could be treated in greater and more beneficial detail.

  1. A) In our opinion, the information within the neurotransmitters section are sufficient for what the rest of the review

R1) 226-8: I do not believe that Eccles, Hodgkin, or Huxley ever worked on Aplysia. H&H won the Nobel for their work in the squid giant axon. Eccles’ principal subjects were the frog NMJ and cat spinal cord in which he and his collaborators developed pioneering intracellular recording techniques. Eric Kandel, however, did receive the Nobel in 2000, largely for his work in Aplysia learning and memory. Is it possible that it is Kandel and the 2000 Nobel the authors are thinking of?

  1. A) The reviewer is right and we apologize for the careless error, which we corrected.

R1) 335 “outer layer” substitute a term such as crust or biosphere or change sentence to better describe what is meant

  1. A) We substituted the term as suggested.

R1) 971 beginning with this paragraph, a number of topics are taken up and disposed of without much connection to preceding sections. Perhaps they could be better organized to constitute a separate section on mixed effects of multiple toxicants and of the influence of abiotic factors on the toxicology of groups of pollutants. I actually think that may have been the intention of the authors, to discuss mixtures and modifying influences, but the material needs to be organized around a cohesive theme.

  1. A) Probably there has been a misunderstanding about the “Omics studies” section. Therefore, this section has been reorganized, firstly describing omics studies considering effects of single species of pollutants in some of the marine invertebrates selected in this review, and secondly describing those omics studies that considered the effects of mixtures of contaminants. Some information not really relevant has been removed.

R1) The entire manuscript would benefit greatly from editing by an English language editor. Nearly every sentence up to line 540 has a problem. Most are quite minor, but cumulatively, the mistakes exact a toll on the reader.

  1. A) English editing was performed on the entire text.

    R1) This manuscript takes on too many subjects and as a result, most subjects are treated somewhat superficially.

    1. A) we extensively modified the manuscript, adding, removing or correcting almost all sections, trying to correct these gaps.

    R1) The metals section that begins on manuscript line 390 is much less successful. Rather than a true review, the authors set the pattern of summarizations of data in the studies they cite, with no or low insights. For example, the first toxicant considered, Cu, could be introduced by alerting a reader that the threshold hemolymph concentration of Cu regarded as external (because many invertebrates use hemocyanin as a respiratory pigment) that causes deleterious effects varies with systematic group and life stage … but there is no such preamble or summary given.

    1. A) The metals section has been extensively revised, and re-organized. New insights have been added throughout the section.

    R1) Furthermore, information is not organized within a topic according to any conventional standard: by life stage, by physiological system affected, by systematic group. Without perspective and insightful organization, there is little advantage to a reader of such a metals review beyond the bibliography.

    1. A) This section is organized in two sub-sections, i.e. essential and non-essential metals, and within each of them we tried to group the data according to the phyla. Information concerning functions within organisms and related neurological pathologies have been added.

    R1) In revision of the metals section, the style of the pesticide and microplastics toxicology sections should be emulated, but even in these latter sections, it is important to summarize with perspective on the main impacts of these classes of pollutants, which is not done to sufficient extent.

    1. A) we added what the reviewer requested.

    R1) Lines 380-382: Here is the goal of this manuscript. The revision should focus more narrowly on this topic and introductory material that serves it. All the introductory material (lines 45-353) should be revamped to reflect the narrower focus.

    1. A) all the introductory sections (preceding lines 380-382 in the first version of the manuscript) have been revised, re-organized and/or re-written according to the reviewer suggestion.

    R1) It is not clear why the superficial phylogeny and systematics of larval and postlarval invertebrate groups (lines 122-272) is included. What is the goal of relating tidbits of invertebrate phyla systematics, the NS design of echinoderms, mollusks and crustaceans, plus a few details of their sensory systems, etc? I believe they are meant as introductory material to the effects of metals and other pollutants on the NS, but almost all of this is completely unnecessary, and covered much better elsewhere in dedicated articles and textbooks on these subjects, to which the authors could refer. These descriptions are not necessary to the later discussions of pollutants that act on neurotransmitters of invertebrate NS.

    1. A) In our opinion, the information included in this paragraph (renamed as “General remarks of marine invertebrates as model organisms and basic notes of their nervous system”) are important to have an immediate view of the main structures of the NS in the different marine invertebrates selected in this review. We are aware that this topic is better covered by dedicated articles and textbooks, to which of course we refer, but we think that the description of the NS is a useful introductory and preparatory part for the following paragraphs of this review. It only serves to recall some notions of the NS of the marine invertebrates described here. Indeed, this review describes the effects of pollutants on nerve regions/structures, affecting both larvae and adults of marine invertebrates (echinoderms, molluscs and crustaceans) and we therefore think it might be useful for a reader unfamiliar with these topics to have a brief description of the NS anatomy, albeit superficial, in this same review. Even the figures, albeit very stylized, are explanatory of the typical nervous structures of the aforementioned invertebrates. However, we added this explanation at the beginning of the section. Besides, we have extensively revised this section too, reducing the descriptions of the NS, eliminating those unnecessary descriptions for the main text, such as the “Neuropeptides” section, which has been completely removed.

    R1) The neurotransmitter section has merit as a lead-in to marine pollutants, but if the other introductory material were eliminated, this subject could be treated in greater and more beneficial detail.

    1. A) In our opinion, the information within the neurotransmitters section are sufficient for what the rest of the review

    R1) 226-8: I do not believe that Eccles, Hodgkin, or Huxley ever worked on Aplysia. H&H won the Nobel for their work in the squid giant axon. Eccles’ principal subjects were the frog NMJ and cat spinal cord in which he and his collaborators developed pioneering intracellular recording techniques. Eric Kandel, however, did receive the Nobel in 2000, largely for his work in Aplysia learning and memory. Is it possible that it is Kandel and the 2000 Nobel the authors are thinking of?

    1. A) The reviewer is right and we apologize for the careless error, which we corrected.

    R1) 335 “outer layer” substitute a term such as crust or biosphere or change sentence to better describe what is meant

    1. A) We substituted the term as suggested.

    R1) 971 beginning with this paragraph, a number of topics are taken up and disposed of without much connection to preceding sections. Perhaps they could be better organized to constitute a separate section on mixed effects of multiple toxicants and of the influence of abiotic factors on the toxicology of groups of pollutants. I actually think that may have been the intention of the authors, to discuss mixtures and modifying influences, but the material needs to be organized around a cohesive theme.

    1. A) Probably there has been a misunderstanding about the “Omics studies” section. Therefore, this section has been reorganized, firstly describing omics studies considering effects of single species of pollutants in some of the marine invertebrates selected in this review, and secondly describing those omics studies that considered the effects of mixtures of contaminants. Some information not really relevant has been removed.

    R1) The entire manuscript would benefit greatly from editing by an English language editor. Nearly every sentence up to line 540 has a problem. Most are quite minor, but cumulatively, the mistakes exact a toll on the reader.

    1. A) English editing was performed on the entire text.

Reviewer 2 Report

This review summarizes the current knowledge of neurobiology of 3 major marine invertebrate phyla, 3 major classes of pollutants, and recent studies of how marine pollutants affect neural structure and function.   The paper reviews the neural organization and neural functions for Echinoderms, Molluscs, and Crustaceans and provides summaries of the pollutants: heavy metals, pesticides, and plastics.  The paper examines pollutants that are most frequently present in marine environments and their neurotoxic potential effects focusing on their neurotoxic effects.  The authors conclude that a deeper understanding of the effects of pollution will come from the integration of different types of molecular data and these data have the potential to reveal the role of pollutants on the marine environment.

The paper very thoroughly examines a broad sweep of scientific literature.  The paper is unique in summarizing the molecular neurobiology of 3 major groups of invertebrates and combining it with summaries of pollutants.  This is an area of intersection between studies of neurotoxicity and invertebrate zoology that has been revealed by genomic studies.  The paper emphasizes that the basics of neural function are similar in all metazoans and an opportunity exists to develop more precise and useful models for studies of neurotoxicity.  The paper will attract a broad range of readers and be well cited.  

The paper is very well organized and the information logically presented.  There are several places where awkward phrasing indicates that the paper needs to be more carefully copy edited by a native English speaker.

The paper stresses that studies on marine invertebrates are relevant.  However, the paper would benefit by a brief, but focussed discussion of how these studies can supplement or supplant studies in vertebrates and provide insights into effects on human health.   I would have liked to see a specific example of a molecular mechanism in which a pollutant alters a common neurobiological processes.  Such an example would emphasize the argument that studies of invertebrates can benefit human health. 

Author Response

R2) There are several places where awkward phrasing indicates that the paper needs to be more carefully copy edited by a native English speaker.

  1. A) English editing was performed on the entire text.

R2) The paper stresses that studies on marine invertebrates are relevant.  However, the paper would benefit by a brief, but focussed discussion of how these studies can supplement or supplant studies in vertebrates and provide insights into effects on human health.   I would have liked to see a specific example of a molecular mechanism in which a pollutant alters a common neurobiological processes.  Such an example would emphasize the argument that studies of invertebrates can benefit human health. 

  1. A) We thank the reviewer for the appreciation she/he made to the review. In the conclusions section, we included a brief discussion on the importance of studies conducted on marine invertebrates in the perspective of safeguarding human health and just an example of human disease (myelin dysfunction) related to the environmental pollutants. We have also added sentences in the various paragraphs of pollutants that emphasize their role in the human health.

Round 2

Reviewer 1 Report

The provided neuroanatomy (lines 122-266) still does not rise to the level of useful, for the following reasons:

  1. No toxicology on NS of larval forms of the 3 phyla is reported, with a few exceptions, lines 444-9, but this concerns cell fate, not neurophysiology or neuroanatomy; 690-2 which is a self-contained toxicity report, and 667-78, which is discussed below.
  2. Few relevant details of metals toxicity that concern ganglia, circuits or pathways, relevant ion channels demonstrated to have defined NS roles, or neurotransmitter receptor-effects are reported. The authors usually simply report tissues in which metals accumulate, or effects on ChE. At times, the proposed relevant metal toxicological effects are listed (e.g. Mn effects on dopamine inhibition of ciliary beat rate, lines 387-90), but when this happens, it turns out the neural targets were not treated in the neuroanatomy section in any way. In this stated example, what is described in the Mn paragraph is all that is needed, since the neuroanatomy provided (lines 205-7; 230-5, but specifically lines 232-3) is too brief to be relevant to the stated Mn toxic effects.

Lines 426-43 summarizing the effects of Zn on AChE represent a good example of a stand-alone treatment that requires no additional NS anatomy. Additionally, text of lines 667-78 is self-contained, and goes so far beyond what is included in the anatomy section as to render the latter unnecessary.

  1. None of the subjects of toxin tissue accumulation, even when the tissue is part of the NS, toxin-induced enzyme inhibition, or effects on gene expression require a neuroanatomy lesson unless physiological or behavioral effects are described at the same time. No such effects are detailed anywhere in the ms.

There are a few additional minor mentions of toxicology on invertebrate NS (lines 751-3; 791-3; 806; 1057-61… that also do not require a neuroanatomy section.

The authors still attempt to cover too much ground. Human effects of metals remain a puzzling digression throughout the piece, e.g. lines 332-333; 414-420; 556-7; 578-81, and should be removed. The human disease sentence of lines 458-461 remains completely irrelevant to the details discussed in the rest of the paragraph. Human effects of Cd (lines 527-534) similarly have little to no relevance to the topic of this review. Since I did not mention human effects in the first review, however, I offer this as a suggestion, not a requirement for publication.

Author Response

Response to reviewer’s comments

R:        The provided neuroanatomy (lines 122-266) still does not rise to the level of useful, for the following reasons: 1. No toxicology on NS of larval forms of the 3 phyla is reported, ………………………….. …………………………… There are a few additional minor mentions of toxicology on invertebrate NS (lines 751-3; 791-3; 806; 1057-61… that also do not require a neuroanatomy section.

A:        we removed the paragraph concerning neuroanatomy, including figures 2,3 and 4, leaving only the section related to acetylcholinesterase as a biomarker of neurotoxicity, as most of the collected studies in this review have been focused on the analysis of its activity.

R:        The authors still attempt to cover too much ground. Human effects of metals remain a puzzling digression throughout the piece, e.g. lines 332-333; 414-420; 556-7; 578-81, and should be removed. The human disease sentence of lines 458-461 remains completely irrelevant to the details discussed in the rest of the paragraph. Human effects of Cd (lines 527-534) similarly have little to no relevance to the topic of this review. Since I did not mention human effects in the first review, however, I offer this as a suggestion, not a requirement for publication.

A:        A second reviewer suggested to provide insights into effects on human health, adding some specific examples. If it is not a great problem for this reviewer, we would like to maintain these sentences.